# Oral Malignant Non-Hodgkin Lymphoma: A Retrospective Single-Center Study

**DOI:** 10.3390/ijerph19052605

**Published:** 2022-02-24

**Authors:** Selene Barone, Caterina Buffone, Martina Ferrillo, Federica Pasqua, Stefano Parrotta, Marianna Salviati, Francesco Bennardo, Alessandro Antonelli

**Affiliations:** School of Dentistry, Department of Health Sciences, Magna Graecia University of Catanzaro, 88100 Catanzaro, Italy; selene.barone@studenti.unicz.it (S.B.); caterina.buffone@studenti.unicz.it (C.B.); martina.ferrillo@studenti.unicz.it (M.F.); federica.pasqua@studenti.unicz.it (F.P.); stefano.parrotta@studenti.unicz.it (S.P.); marianna.salviati@studenti.unicz.it (M.S.); alessandro.antonelli@studenti.unicz.it (A.A.)

**Keywords:** lymphoma non-Hodgkin, mouth, oral neoplasms

## Abstract

This study aimed to retrospectively evaluate the incidence of oral non-Hodgkin lymphoma (NHL) in patients referred to the Academic Hospital of the Magna Graecia University of Catanzaro from 2002 to 2020. A retrospective single-center study was performed. Patients with a histologically confirmed diagnosis of oral NHL were included. Demographic data and clinical parameters were digitally recorded, focusing on the NHL-specific localization and symptomatology. The study sample was evaluated by analyzing descriptive statistics with absolute and relative frequencies. A total of 26 patients with intraoral NHL were identified with a progressive increase in NHL occurrence during the observation period. Clinical manifestations included swelling/mass (80.7%), eventually associated with pain and ulcerations. The most common localizations were in soft tissues: buccal mucosa (38.4%), tongue (19.2%), gingiva (11.5%), cheek (11.5%). Oral NHL is rare. Clinical manifestations were unspecific, so a misdiagnosis could occur. The extranodal B-cell form of oral NHL, particularly diffuse large B-cell lymphoma, was the most common frequent oral NHL in this southern Italian population, with a progressively increased occurrence in almost 20 years.

## 1. Introduction

Lymphomas are a malignant neoplastic proliferation of the immune system and represent the second most common primary malignancy in the head and neck [1]. They are classified in Hodgkin-lymphoma (HL) and non-Hodgkin-lymphoma (NHL). NHL incidence is rising in many countries; in the head and neck area, it varies from 1% to 17% [2]. From 1994, based on the Revised European-American Classification of Lymphoid Neoplasm (REAL), lymphomas are currently classified using the World Health Organization (WHO) classification that recognizes over 20 different subtypes of NHL and consider morphology, immunophenotype, genetic and clinical features [3,4,5]. Between 85% and 90% of all NHL derive from B lymphocytes, and only 10% arise from T lymphocytes or natural killer cells [6,7]. Most lymphomas show heterogeneous extranodal manifestations [7,8]. Depending on the specific subtype, NHL can involve Waldeyer’s tonsillar ring, the major salivary glands, and paranasal sinuses [7,9,10]. Oral cavity is an uncommon site for NHL, representing only 2% of all extranodal lymphomas. Intraoral NHL can occur in jaw bones, although soft tissues, such as palatal mucosa, gingiva, tongue, cheek, the floor of the mouth and lips are the most common sites [11,12,13,14,15]. Lymphomas of the oral cavity can appear as dental abscesses, an epithelial tumor, or other diseases, such as medical-related osteonecrosis of the jaws [9,10,16,17,18]. The etiology of NHL is still unknown [2]. However, patients exposed to pesticides and radiation or affected by autoimmune diseases, such as rheumatoid arthritis, systemic lupus erythematosus, and the Sjögren syndrome viral infection with Epstein-Barr virus (EBV), human T-cell lymphotropic virus 1 (HTLV-1), human immunodeficiency virus (HIV), human herpes virus type 8 (HHV-8), hepatitis B and C virus (HBV and HCV), have been related to a greater risk of NHL [19,20].

This study aimed to retrospectively evaluate the incidence of oral NHL in patients referred to our clinic from January 2002 to December 2020.

## 2. Materials and Methods

### 2.1. Study Sample

This study was designed as an observational, descriptive, and retrospective single-center study. All patients with oral mass and a primary diagnosis of oral lymphoma presenting between January 2002 to December 2020 at the Oral Pathology Unit of Academic Hospital of Magna Graecia University of Catanzaro, Italy, were included in the study. Inclusion criteria were (1) histologically confirmed diagnosis of lymphoma, (2) location in the oral cavity, and (3) complete clinical information. According to the Declaration of Helsinki on medical protocol and ethics, the regional Ethical Review Board of Central Calabria (reference for the Magna Graecia University of Catanzaro) approved the study (189/2019).

### 2.2. Data Collection Method and Study Variables

Epidemiologic data were digitally recorded, focusing on demographic and lifestyle/occupational data (exogenous factors, such as chemicals and agricultural exposures), clinical parameters, and the medical history of each patient. Recorded variables included age at diagnosis, gender, clinical features of the mass, specific localization in the oral cavity, symptoms (i.e., B-symptoms: fever greater than 38 °C; drenching night sweats; unintentional weight loss of at least 10% of their body weight over six months or fewer), and signs (i.e., tooth mobility and mucosal bleeding), HIV and HBV positivity status. Pain levels were evaluated through a visual analog scale (VAS), only in patients with pain. An incisional biopsy was performed in all patients with a clinical suspicion of oral lymphoma. Histologic specimens were classified according to REAL up to 2008 and WHO classification since 2008 [5]. All patients also underwent contrast CT scans of the head and neck, chest, abdomen, and pelvic cavity. The most relevant data of radiological evaluation were collected. Treatment modality and five-year survival data were recorded only for patients treated in the Academic Hospital of Magna Graecia University of Catanzaro. Data were not available for patients treated after diagnosis in other hospitals.

### 2.3. Statistical Analysis

Data were collected through Microsoft^®^ Excel (Microsoft Office, Microsoft Corporation, Redmond, WA, USA) and subsequently analyzed using the STATA software program (STATA 11, StataCorp, College Station, TX, USA). The study sample was evaluated by analyzing descriptive statistics with absolute and relative frequencies.

## 3. Results

Patients’ data are reported in Table 1 and a summary of patients’ characteristics is reported in Table 2.

### 3.1. Study Sample

From 2002 to 2020, a total of 115 patients were diagnosed with head and neck NHL in the Academic Hospital of Magna Graecia University of Catanzaro. Of these, 26 patients were diagnosed with oral NHL in the Oral Pathology Unit, and were included in this study. The mean age was 58 years. Fifteen patients were male (57.6%), and 11 were female (42.3%), with a male to female ratio of 1.36:1. Regarding lifestyle and occupational exposure, 15 patients were exposed to risk factors (57.7%), 4 were not exposed (15.4%) and data were not available for 7 patients (26.9%). Data analysis showed a progressive increase in the diagnosis of oral NHL (Figure 1): from 2002 to 2006, NHL was diagnosed in 4 patients (15.3%); from 2007 to 2011, 4 patients (15.3%) received a diagnosis of NHL; from 2012 to 2016, NHL was diagnosed in 7 patients (26.9%); from 2017 to 2020, NHL was identified in 11 patients (42.3%).

### 3.2. NHL Manifestations

According to clinical localization, extranodal NHL involved both hard and soft tissues. Soft tissues were the most common sites of disease (80.7%): 10 in the buccal mucosa (38.4%; Figure 2), 5 in the tongue (19.2%), 3 in the gingiva (11.5%), 3 in the cheek (11.5%; Figure 3). Hard tissue involved were palate (4 patients, 15.4%) and mandible (1 patient, 3.8%; Figure 4 and Figure 5).

A computed tomography scan of a bone site of lymphoma could show an irregular density pattern (Figure 5). The most common clinical presentations were swelling or mass gradually increasing in size (21 patients, 80.7%). Ulcerations were found in 13 patients (50%). The pain was reported in 6 patients (23%) with a mode VAS pain score (scale 0–10) of 5. Other symptoms/signs included paresthesia, difficulties in swallowing, tooth mobility, and mucosal bleeding. Serological analysis highlighted 5 HIV+ patients (19.2%), and 6 EBV+ patients (23%). Histopathological analysis confirmed the diagnosis of NHL for all patients. The most common histologic subtype was diffuse large B-cell lymphoma (11 patients, 42.3%), followed by plasmablastic lymphoma (7 patients, 26.9%), extranodal marginal zone lymphoma of mucosa-associated lymphoid tissue (MALT) (2 patients, 7.6%), peripheral T-cell lymphoma (2 patients, 7.6%), Burkitt lymphoma (1 patient, 3.8%), follicular lymphoma (1 patient, 3.8%), small lymphocytic lymphoma (1 patient, 3.8%) and mantle cell lymphoma (1 patient, 3.8%). Regarding treatment, 10 patients received chemotherapy (38.4%), 8 received a combination of chemotherapy and radiotherapy (30.8%) and data were not available for 8 patients (30.8%). Five-year survival data were available for only 18 patients (11 alive, 7 died).

## 4. Discussion

Oral NHLs are usually diagnosed by biopsy of a mass or enlarged lymph node [21,22]. The head and neck are the second most common sites of extranodal non-Hodgkin lymphoma, after the gastrointestinal tract [8,23,24]. The more frequent locations of malignant lymphoma are Waldeyer’s tonsillar ring and cervical lymph nodes, followed by the nose, paranasal sinuses, orbits, and salivary glands [25]. Although about 90% of all malignancies are squamous cell carcinomas and primary lymphoma of the oral cavity is very rare, representing only 3% of all lymphomas in the general population, the incidence of the disease is increasing [26,27]. Swelling or pain is the most common initial complaint among most patients with oral NHL [21,22]. This report aimed to investigate the occurrence of oral NHL, analyzing localization, symptoms/signs, clinical and histopathological features in patients presented at the Academic Hospital of the Magna Graecia University of Catanzaro from 2002 to 2020. This retrospective study included 26 patients diagnosed with oral NHL, confirmed by histopathological analysis. According to relevant European and North American studies, the mean age of this Italian population was 58 years, and higher if compared with South-African populations [9,11,12,13,14,22,28]. European and North American countries have an older population with a longer life expectancy than Africa, influencing the NHL incidence with diet and lifestyle, both identified as etiological factors for non-Hodgkin lymphomas [29]. According to literature, NHL distribution highlighted a mild male prevalence, with a male to female ratio of 1.36:1 [2,14,21,22,23,24,25,26,27,28,29,30]. The diagnosis of oral NHL was rare in our center, considering 26 cases encountered in the 18 years examined. In a cross-sectional study with a larger sample size, Iguchi and colleagues stated that lymphomas represent 14% of the malignancies in the head and neck region. Of these, 97% are NHL, with a high percentage of extranodal forms [2]. The intraoral localization of NHL could include both hard and soft tissues, such as gingiva, tongue, buccal mucosa, cheek, oral mouth floor, lips, palate and bones [9,10,14,30,31,32]. In this retrospective study, soft tissues were more commonly involved than hard tissues by NHL, with greater involvement of buccal mucosa, followed by rare localizations, such as the tongue, gingiva and cheek. Tseng et al. have recently reviewed 607 cases of oral lymphomas: concerning location, they found that the most affected site was the gingiva, followed by the palate, maxilla, mandible, tongue, buccal mucosa, vestibule, lip and floor of the mouth [33]. The rare incidence and the common signs and symptoms in the oral cavity determine important factors for the misdiagnosis of NHL [26,34,35]. Regarding clinical manifestations, the diagnosis of oral NHL is also very complicated for the heterogeneity of clinical and radiographic signs. An osteolytic lesion with an ill-defined border is the most frequent radiographic presentation of NHL occurring in the hard tissue. These radiological images could be similar to osteomyelitis [36]. Tseng et al. reported some cases of oral lymphoma presented as a periapical radiolucency mimicking an endodontic lesion [32]. Oral NHL usually causes pain, discomfort associated with local swelling, ulcerations, gingival edema or tooth mobility due to alveolar bone loss [24,26,37]. We observed a mode VAS pain level of 5, corresponding to moderate pain on this descriptive pain-intensity scale. These results suggested future investigation regarding pain management [38]. An altered lip sensitivity or pathological fractures could occur after malignancy progression, determining the involvement of the jaws [26]. According to the literature, this cohort of patients showed unspecific signs and symptoms. Regarding extraoral features, swelling/mass was the most frequent clinical manifestation, present in almost all patients, with or without pain. Likewise, the intraoral examination did not record significant signs. However, oral NHL manifestations were usually determined by mucosal ulceration, followed by tooth mobility and mucosal bleeding. Paresthesia and difficulties in swallowing often occurred without specific indications of injury. Unfortunately, these clinical signs are non-specific and could be a confounding factor in the diagnosis. Specifically, they can be related to endodontic infection, periodontitis, osteomyelitis, autoimmune diseases and other malignant lesions [39,40]. For these reasons, incorrect treatment plans often delay the diagnosis of oral lymphoma [26]. NHL affects 4% to 10% of patients with HIV disease. The etiology is still unknown, but the recent increase in incidence may be related to immunodeficiency conditions. Undoubtedly, immunodeficiency-HIV-related patients have an increased incidence of oral NHL for their systemic complications [26,33]. However, in the southern Italian regions and rural areas, the low risk of HIV infection reduces the incidence of immunodeficiency-related NHL, as reported in this study. In our center, among these known oncogenic viruses, the EBV was found in six patients and HIV in five patients. Other patients were suffering from other causes of immunosuppression. Diagnosis of NHL is based on histologic specimen examination, and it is commonly accepted as a predominantly B-cell lineage [21]. The most common histopathological subtypes were diffuse large B-cell lymphoma and plasmablastic lymphoma. Diffuse large B-cell lymphoma has an increased prevalence in elderly patients, occurring on average in the seventh decade of life. This histological type may involve both the hard and soft tissue with a non-specific clinical appearance in the oral cavity, most commonly presenting as a painless swollen mass [41]. Plasmablastic lymphoma, originally described as a disease that involves the oral cavity of immunodeficient patients, is an aggressive and rare diffuse large B-cell lymphoma subtype. It is usually associated with a poor prognosis and low survival rate after diagnosis (4–11 months) [42]. Most of the patients (78.9%) included in this study were exposed to occupational factors (exogenous factors, such as chemicals and agricultural exposures). These results are in line with literature [43]. According to the data reported here, the use of chemotherapy or chemotherapy concomitantly with radiotherapy had a favorable outcome [44]. In this study, the 5-year overall survival rate was 61.1%.

The main limit of this study is the design as a descriptive and retrospective single-center study of a population presented at a dedicated unit of dentistry and oral surgery. Moreover, the study population may not reflect the oral NHL prevalence in other regions of southern Italy.

## 5. Conclusions

Although NHL is highly uncommon in the oral cavity, our results highlighted an increase in oral NHL, in daily practice, over almost 20 years in a local south Italian population. To limit misdiagnoses and diagnostic delay of oral NHL, unspecific signs of mucosal ulcerations, bleeding, and tooth mobility require careful evaluation. Following the literature, the B-cell form of oral NHL was more frequent, and its clinical manifestations included swelling eventually associated with pain and discomfort. Further studies are needed to define NHL characteristics and occurrence in the oral cavity.

## Figures and Tables

**Figure 1 ijerph-19-02605-f001:**
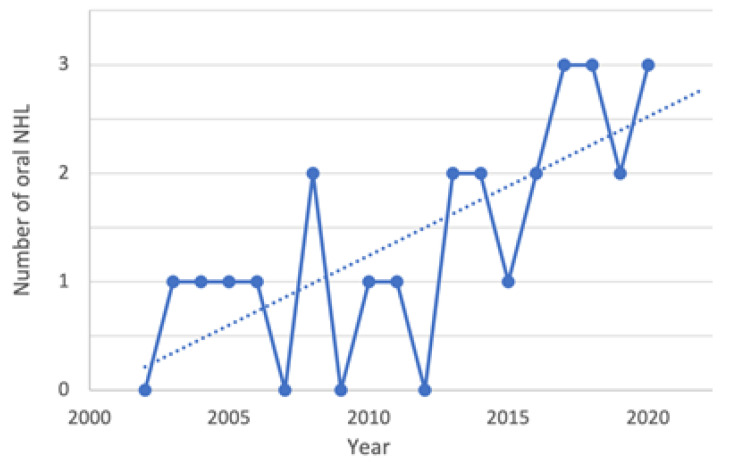
Graphic evaluation of the number of oral NHL cases per annum from 2002 to 2020. During the observation period, a progressive increase in the incidence emerged.

**Figure 2 ijerph-19-02605-f002:**
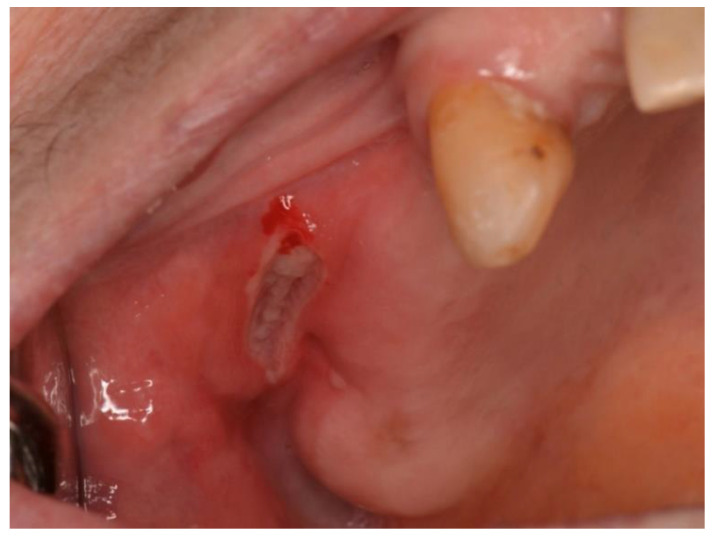
Mucosal ulceration associated with pain in a patient affected by a B-cell non-Hodgkin lymphoma.

**Figure 3 ijerph-19-02605-f003:**
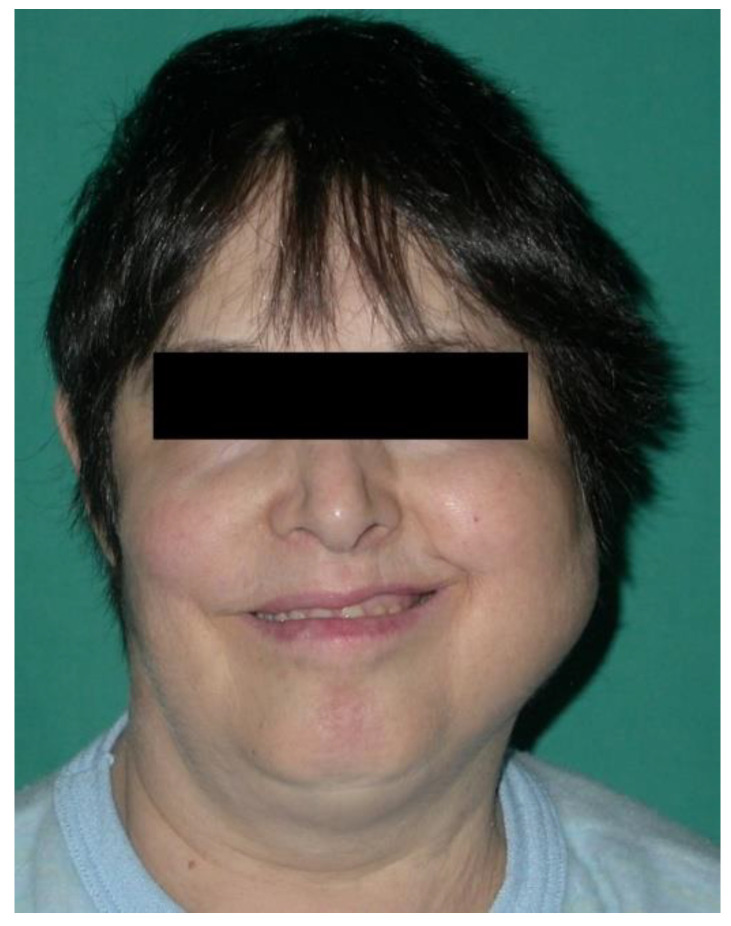
Extraoral evaluation of patients with an aggressive B-cell non-Hodgkin lymphoma of the left cheek. Clinical features: significative swelling, limited mouth opening movement associated with pain.

**Figure 4 ijerph-19-02605-f004:**
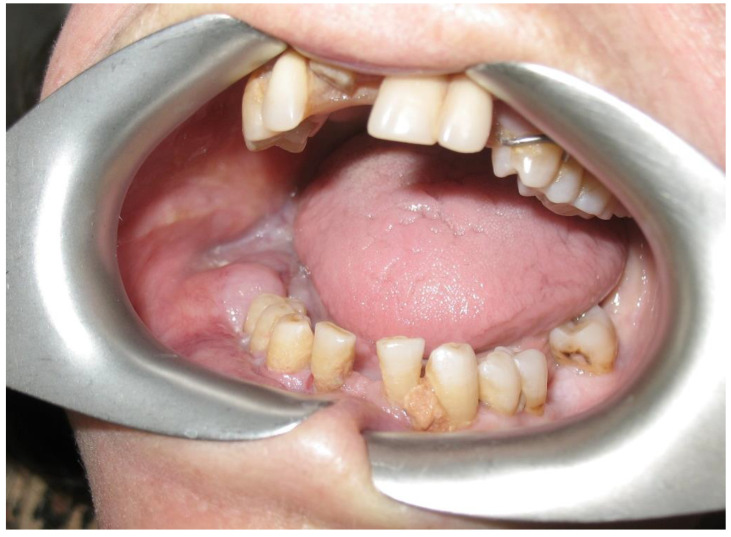
Intraoral evaluation of a patient with a B-cell non-Hodgkin lymphoma of the mandible mimicking a dental abscess. Clinical manifestations included mucosal swelling, tooth mobility, and numb chin syndrome.

**Figure 5 ijerph-19-02605-f005:**
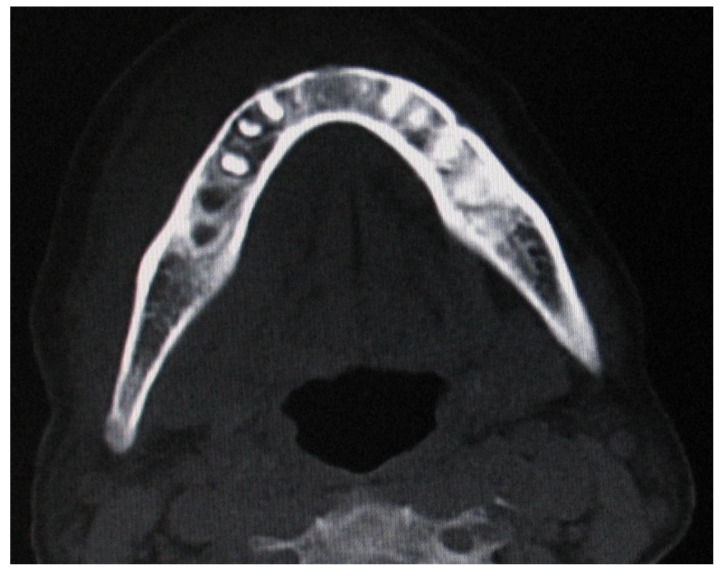
Computed tomography scan of a patient with a B-cell non-Hodgkin lymphoma of the mandible mimicking a dental abscess showed an irregular density pattern.

**Table 1 ijerph-19-02605-t001:** Patients’ data.

Case	Year of Diagnosis	Gender	Age (Years)	Occupational Exposure	Site	Signs and Symptoms	VAS	Diagnosis	Serological Test	Treatment/
1	2003	M	68	N/A	Cheek	Swelling, pain	5	DLBCL	EBV+	Outcome
2	2004	M	46	N/A	Palate	Swelling, ulceration	-	BL	-	CT + RT/died
3	2005	F	52	Yes	Cheek	Swelling, mucosal bleeding	-	DLBCL	-	N/A
4	2006	M	56	Yes	Gingiva	Swelling, ulceration	-	PBL	HIV+	CT + RT/alive
5	2008	F	66	Yes	Tongue	Swelling, pain	2	DLBCL	-	N/A
6	2008	F	82	N/A	Palate	Swelling, ulceration	-	SLL	-	CT/alive
7	2010	M	45	Yes	Tongue	Swelling, ulceration	-	DLBCL	EBV+	N/A
8	2011	M	42	Yes	Buccal mucosa	Mucosal bleeding, difficulties in swallowing	-	ENMZL	EBV+	CT/alive
9	2013	M	49	N/A	Buccal mucosa	Swelling, ulceration	-	PBL	HIV+	N/A
10	2013	F	51	Yes	Buccal mucosa	Mucosal bleeding, ulceration	-	PBL	HIV+	CT/alive
11	2014	M	55	No	Tongue	Swelling, difficulties in swallowing	-	PBL	HIV+	CT + RT/alive
12	2014	F	73	N/A	Buccal mucosa	Swelling	-	DLBCL	-	N/A
13	2015	M	43	Yes	Cheek	Swelling, ulceration	-	DLBCL	-	CT + RT/died
14	2016	F	61	Yes	Palate	Swelling, ulceration	-	PTCL	EBV+	CT + RT/alive
15	2016	F	76	No	Buccal mucosa	Pain, ulceration	5	DLBCL	-	CT/died
16	2017	M	45	N/A	Gingiva	Bleeding	-	FL	-	CT + RT/alive
17	2017	M	72	Yes	Buccal mucosa	Paresthesia	-	ENMZL	EBV+	N/A
18	2017	M	54	Yes	Tongue	Swelling	-	DLBCL	-	N/A
19	2018	F	70	No	Gingiva	Swelling, pain, ulceration	3	PBL	-	CT + RT/alive
20	2018	F	60	N/A	Mandible	Swelling, pain, tooth mobility, paresthesia	5	DLBCL	-	CT/died
21	2018	M	58	Yes	Buccal mucosa	Swelling, ulceration	-	MCL	-	CT + RT/alive
22	2019	M	53	Yes	Buccal mucosa	Swelling	-	PBL	HIV+	N/A
23	2019	M	64	Yes	Tongue	Swelling, difficulties in swallowing	-	DLBCL	EBV+	CT/died
24	2020	M	44	Yes	Buccal mucosa	Swelling	-	PBL	-	CT/alive
25	2020	F	74	No	Buccal mucosa	Swelling, ulceration	-	DLBCL	-	CT/died
26	2020	F	51	Yes	Palate	Swelling, pain, ulceration	4	PTCL	-	CT/alive

Legend: DLBCL: diffuse large B-cell lymphoma; PBL: plasmablastic lymphoma; ENMZL: extranodal marginal zone lymphoma of mucosa-associated lymphoid tissue (MALT); PTCL: peripheral t-cell lymphoma: BL: Burkitt’s lymphoma; FL: follicular lymphoma; SLL: small lymphocytic lymphoma: MCL: mantle cell lymphoma; N/A: Not available; CT: Chemotherapy; RT: Radiotherapy.

**Table 2 ijerph-19-02605-t002:** Summary of patients’ characteristics.

Patients	*n*	26	100%
Gender	Male	15	57.6%
	Female	11	42.3%
Age (years)	Mean (range)	58 (42–82)	
Occupational exposure	Yes	15	57.7%
	No	4	15.4%
	Not availlable	7	26.9%
Site	Buccal mucosa	10	38.4%
	Tongue	5	19.2%
	Gingiva	3	11.5%
	Cheek	3	11.5%
	Palate	4	15.3%
	Mandible	1	3.8%
Signs and symptoms	Swelling	21	80.7%
	Ulceration	13	50%
	Pain (VAS mode)	6 (5)	23%
	Paresthesia	2	7.6%
	Difficulties in swallowing	3	11.5%
	Tooth mobility	1	3.8%
	Mucosal bleeding	4	15.3%
Diagnosis	DLBCL	11	42.3%
	PBL	7	26.9%
	ENMZL	2	7.6%
	PTCL	2	7.6%
	BL	1	3.8%
	FL	1	3.8%
	SLL	1	3.8%
	MCL	1	3.8%
Serological test	EBV+	6	23%
	HIV+	5	19.2%
Treatment	CT	10	38.4%
	CT + RT	8	30.8%
	Not availlable	8	30.8%
Outcome	Alive	11	42.3%
	Died	7	26.9%
	Not availlable	8	30.8%

Legend: DLBCL: diffuse large B-cell lymphoma; PBL: plasmablastic lymphoma; ENMZL: extranodal marginal zone lymphoma of mucosa-associated lymphoid tissue (MALT); PTCL: peripheral t-cell lymphoma: BL: Burkitt’s lymphoma; FL: follicular lymphoma; SLL: small lymphocytic lymphoma: MCL: mantle cell lymphoma; CT: Chemotherapy; RT: Radiotherapy.

## Data Availability

The data presented in this study are available on request from the corresponding author.

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
