# Peer review of "Oral Malignant Non-Hodgkin Lymphoma: A Retrospective Single-Center Study"

_ijerph, 2022, doi:10.3390/ijerph19052605_

Round 1
Reviewer 1 Report
A reduction in the length of both introduction and discussion praragraph should be done, expecially eliminating the description of Hodgkin versus non Hodgkin's lymphoma, as the paper concern only NHL
Also a summary table on pt characteristics is needed
Patients with myeloma should not be included (2 described in table 1)
The incidence of plasmabalstic lymphoma, in HIV-negative partients is higher than expected, could this be explained somehow ? And possibly compared with the literature
The total number of patients with head & neck NHL seen at the Center should be described in order to show the proportion of cases with oral lymphoma among alla H&N
The figure 1 reflects the increased incidence of oral lymphoma at the single center: does this reflect an i increased incidence of all NHLs at the Center or a special increase of only oral lymphoma? Again , a proportion with other NHL could be interesting
Finally , treatment approach and oputcome should be at least mentioned
Author Response
We thank the reviewer for the careful analysis of our manuscript and for the precious suggestions that significantly aid in improving the quality of our study.
We revised the manuscript following the recommendations in order to enhance article readability. Changes are highlighted in yellow.
Comments to the Author:
- A reduction in the length of both introduction and discussion paragraph should be done, expecially eliminating the description of Hodgkin versus non Hodgkin's lymphoma, as the paper concern only NHL
- Also a summary table on pt characteristics is needed
- Patients with myeloma should not be included (2 described in table 1)
- The incidence of plasmabalstic lymphoma, in HIV-negative partients is higher than expected, could this be explained somehow ? And possibly compared with the literature
- The total number of patients with head & neck NHL seen at the Center should be described in order to show the proportion of cases with oral lymphoma among alla H&N
- The figure 1 reflects the increased incidence of oral lymphoma at the single center: does this reflect an increased incidence of all NHLs at the Center or a special increase of only oral lymphoma? Again , a proportion with other NHL could be interesting
- Finally, treatment approach and oputcome should be at least mentioned
We followed all comments and suggestions reported in the revision.
- We reduced the length of both introduction and discussion sections, specifically deleting the description of Hodgkin's lymphoma to emphasize the description of non-Hodgkin's lymphoma alone.
- Thank you for your suggestion regarding the lack of this information. We have now added details regarding the demographic and pathological characteristics of patients (Table 2).
- Following your suggestions, we have excluded patients with myeloma (4).
- We have modified the information’s about the “serological test” section of Table 1. Unfortunately, due to a typing error in Table 1, we reported errors on the serological status of patients 9 and 11, who were not EBV+, but HIV+. We revised the manuscript correcting this data in Table 1 and Results However, other patients were suffering from other causes of immunosuppression. Our results are in line with previous findings according to which oral lymphomas occur more frequently in patients with HIV infections and, specifically, PBL it also occasionally affects patients with other causes of immunosuppression. Thank you for reporting this inappropriateness.
- In the revised version of our manuscript, in Results section we now report the information concerning the total number of patients with head & neck NHLs analyzing the database of our center, the Academic Hospital of Magna Graecia University of Catanzaro, Italy, in order to show the proportion of cases with oral NHLs among all H&N cases.
- The figure 1 reflects the increased incidence of oral NHL at the single center. The proportion with cases of head & neck NHLs is now shown in
- Considering the precious comments of the reviewer, we reported treatment modality and five-year survival data in Table 1, Table 2, Material and methods, Results and Discussion sections (only for patients treated in our center, with available data) and we also compared our data with literature.
Thanks for your attention in reading this revised version of our manuscript.
Reviewer 2 Report
This study focuses on the incidence of oral non-Hodgkin lymphoma in patients referred in an academic hospital in Italy. While the number of patients is low, this study ranges from 2002 to 2020. Because of the rarity of these pathologies, this study provides elements of interest to better understand the symptomatology of these tumors.
The study is well written and pleasant to read. However, I would have suggested several types of improvements:
- Why in Table 1 VAS was not reported. No pain ? Or no data available ?
- Please add suitable radiological acquisition to show a typical madible lesion for example. Although it is difficult to generalize from such a small number of patients, knowledge of the clinical and radiological manifestations is very important.
- The title must be revised. As it stands it is not a cohort. In a cohort patients does not have pathology during inclusion and potentially develop it during follow-up (the exposures are recorded).
- The evaluation of the causes of the development of these tumors is poorly reported. The only cause mentioned is the infectious status (EBV, HIV) but no information is given on the exposome of patients (lifestyle, occupation ...). This is lacking.
- Line 86 : " NHL distribution was found (p>0.01)". Why p>0.01, no mistake ?
Author Response
We want to thank the reviewer for the positive evaluation, the comments and the suggestions. We revised the original version of our manuscript following the recommendations in order to enhance article readability. Changes are highlighted in yellow.
Comments to the Author:
This study focuses on the incidence of oral non-Hodgkin lymphoma in patients referred in an academic hospital in Italy. While the number of patients is low, this study ranges from 2002 to 2020. Because of the rarity of these pathologies, this study provides elements of interest to better understand the symptomatology of these tumors.
The study is well written and pleasant to read. However, I would have suggested several types of improvements:
- Why in Table 1 VAS was not reported. No pain? Or no data available?
- Please add suitable radiological acquisition to show a typical mandible lesion for example. Although it is difficult to generalize from such a small number of patients, knowledge of the clinical and radiological manifestations is very important.
- The title must be revised. As it stands it is not a cohort. In a cohort patients does not have pathology during inclusion and potentially develop it during follow-up (the exposures are recorded).
- The evaluation of the causes of the development of these tumors is poorly reported. The only cause mentioned is the infectious status (EBV, HIV) but no information is given on the exposome of patients (lifestyle, occupation ...). This is lacking.
- Line 86: " NHL distribution was found (p>0.01)". Why p>0.01, no mistake ?
We followed all comments and suggestions reported in the revision.
- As specified in the materials and methods, we evaluated pain level through a visual analog scale (VAS). VAS values were considered only in the 6 patients presenting with pain. In table 1, VAS was not reported in the 20 pain-free patients. In fact, in these 20 patients, in table 1, the section "signs and symptoms" does not report pain, but others such as swelling, difficulties in swallowing or ulceration.
Following your suggestions, we have now revised the manuscript specifying, in materials and methods, that the mode VAS pain score was calculated only in the 6 patients with pain.
- Regarding the lack of a radiological documentation, we agree with the reviewer comment. Thank you for noticing this missing information. We have now added, in figure 5, details regarding the three-dimensional radiological characteristics showing an irregular radiolucent pattern of a B-cell non-Hodgkin lymphoma of the mandible
- We agree with the reviewer comment. We improved the content of the title to better reflect the aim and design of our study.
- Thank you for this precious comment. We have now revised the manuscript integrating information on the causes of the development of non-Hodgkin's lymphoma in patients of our center. Please find these details in the table 1, results and discussion
- Thank you for noticing this incorrect information. Mistake. We have now corrected this.
Thanks for your attention in reading this revised version of our manuscript.